# A Study of Electroplated Nanoporous Copper Using Aberration-Corrected Transmission Electron Microscopy

**DOI:** 10.3390/nano14060487

**Published:** 2024-03-08

**Authors:** Jianqiang Wang, Jintao Wang, Ziwen Lv, Luobin Zhang, Fengyi Wang, Hongtao Chen, Mingyu Li

**Affiliations:** 1Department of Materials Science and Engineering, Harbin Institute of Technology (Shenzhen), Shenzhen 518055, China; wangjianqiang@stu.hit.edu.cn (J.W.); lzwen_mail@163.com (Z.L.); 19b955001@stu.hit.edu.cn (L.Z.); 22b955001@stu.hit.edu.cn (F.W.); myli@hit.edu.cn (M.L.); 2State Key Laboratory of Advanced Solder and Joining, Harbin Institute of Technology, Harbin 150001, China; 3Sauvage Laboratory for Smart Materials, Harbin Institute of Technology (Shenzhen), Shenzhen 518055, China

**Keywords:** electroplating, etching, nanoporous copper, capillary force

## Abstract

Nanoporous Cu foam is widely applied in many fields such as the packaging of electronic power devices. In this study, a sandwich-structured Cu-Zn eutectic alloy precursor composed of Cu_0.53_Zn_0.47_/Cu_5_Zn_8_/Cu_0.53_Zn_0.47_ is prepared through electroplating. The surface layer of the precursor, Cu_0.53_Zn_0.47_, has a flat surface with numerous grain boundaries, which effectively promotes its dealloying behavior. By contrast, Cu_5_Zn_8_ has a porous structure, which promotes the dealloying behavior at the center of the precursor. The dealloying of Cu_0.53_Zn_0.47_ is dominated by the coherent surface diffusion of Cu atoms, and the crystal lattice and orientation show no changes before and after dealloying. By contrast, the dealloying behavior of Cu_5_Zn_8_ requires the renucleation of Cu crystals; in this process, Cu atoms are transported to the surface of the layer by capillary forces to form clusters, which nucleate and grow.

## 1. Introduction

Nanoporous metals are generally considered to be a type of porous metal with nanosized ligaments and pores as well as open-cell structures, and have been applied in various fields, such as catalysis [1], batteries [2,3], and sensing [4,5]. In recent years, with the development of electronic components in the direction of smaller size, higher performance, and lower energy consumption, traditional chip interconnection technology has been severely challenged, and electronic packaging researchers have begun to use nanoporous metal as a new interconnection materials.

In advanced 2.5D/3D packaging technology, the pitch size of Cu pillars is less than 20 μm [6]. Soldering can no longer meet the needs of such micro-joints, so all-Cu interconnect technology for next-generation interconnect nodes has been developed. Nanometal materials are used in all-Cu interconnect technology because of their nanosize effect, which can significantly reduce the melting point [7]. Cu nanopaste made of nano-Cu particle materials mixed with organic additives was first studied [8,9,10]. Organic additives can reduce nanoparticle aggregation and resist particle oxidation, but they also lead to a large number of voids in the joint. Voids reduce the physical properties and reliability of the joint. Cu nanopaste needs to be printed on the surface of the pad, and nanoporous copper is a self-supporting material that can be directly prepared on the surface of the pad. Therefore, the use of organic additives can be avoided during the sintering process and the density of the sintered joint can be improved [11]. Dimitrov et al. proposed a method of plating a layer of Sn on the surface of nanoporous Cu [12]. This method can reduce the sintering temperature and improve oxidation resistance. The sintering temperature of their Cu-on-Cu joint only needs to be 200 °C.

Nanoporous copper is usually prepared using the dealloying method, which can selectively remove the less noble metal elements from alloys and leave the more noble metal elements, resulting in the formation of a porous structure. Hayes et al. prepared a precursor alloy through smelting, and then, dealloyed it to prepare uniform nanoporous Cu sheets [13]. Nanoporous copper sheets exhibit brittleness and are prone to fracture during bending or stretching, which limits their use in interconnection applications. Thus, a new approach for preparing nanoporous Cu films on the surface of substrate is proposed. The precursor alloy is obtained through electrodeposition instead of the melting method. Currently, the main process for depositing precursor alloy layers is Cu-Zn electroplating. The dealloying process has a significant impact on the size of the ligament. Mohan et al. prepared nanoporous copper through chemical dealloying, and its ligament size was about 60 nm [11]. Castillo et al. prepared nanoporous copper through electrochemical dealloying, and its ligament size was smaller at 20–33 nm [14].

The traditional Cu-Zn electroplating process uses cyanide as a complexing agent. However, cyanide is highly toxic and environmentally unfriendly. Therefore, various cyanide-free Cu-Zn electroplating processes have been developed. According to the primary complexing agent used, the types of Cu-Zn electroplating are classified as pyrophosphate, citrate, ethylenediaminetetraacetic acid (EDTA), glycerol, and 1-hydroxyethyl diphosphonic acid (HEDP), among others. In our work, pyrophosphate was used as a complexing agent. Pyrophosphate has a coordination effect on both Cu^2+^ and Zn^2+^, and the deposition overpotential of Cu^2+^ is very large in this bath, which is beneficial to the co-deposition of Cu and Zn. In practical applications, Cu-Zn electroplating is often applied in imitation gold electroplating, and the Zn content of the coating is about 70%. Dimitrov et al. believe that the content of the Zn element in a Cu-Zn coating should be above 60 at% to obtain high-quality nanoporous copper [15]. Therefore, in order to obtain a high-Zn precursor alloy layer, the traditional Cu-Zn electroplating solution formula needs to be optimized.

In this study, we aim to enhance the Cu-Zn coupling effect by adding macromolecules to a plating solution to obtain a Cu-Zn coating with a uniform distribution of Zn elements. This coating is then chemically etched to produce a nanoporous Cu coating. Subsequently, the atomic-level mechanisms of the precursor-formation and etching behaviors of nanoporous Cu are investigated using spherical aberration-corrected transmission electron microscopy (AC-TEM).

## 2. Experimental

First, a Cu-Zn alloy layer was electroplated on the surface of a Cu substrate using a pyrophosphate-plating solution containing a composite additive. The plating solution included precursors, complexing agents, acid–base modifiers, and additive packages. The pyrophosphate system of the electroplating solution contained *N*,*N*,*N*′,*N*′-tetrakis(2-hydroxypropyl)ethylenediamine, histidine, sodium saccharin, and low-ester pectin.

The complexing agent was 330 g/L potassium pyrophosphate, and the precursors were 12.5 g/L CuSO_4_·5H_2_O and 43.1 g/L ZnSO_4_·7H_2_O. The pH of the solution was adjusted to 12.5 by using KOH, and the temperature of the plating solution was 50 °C. The molar ratio of Cu^2+^:Zn^2+^ in the plating solution was 1:2, and 500 mL of the plating solution was added to a square plating bath. The anode was an H59 brass plate, the Cu substrate was the cathode, and the ratio of the anode area to the cathode area was 2:1. The duty cycle of the pulsed power supply was 80%, the frequency was 500 Hz, and the average cathodic current density was 6 A/dm^2^. Magnetic stirring was performed at 600 rpm. The plating time was set to 4 h. The coating obtained after electroplating was annealed at 250 °C to release the stress.

Electrochemical tests were performed using a Shanghai ChenHua CHI760E electrochemical workstation, and the experiments were conducted using a three-electrode system. The reference electrode was a Hg/HgO electrode, the counter electrode was a 1 cm × 1 cm Pt electrode, and the working electrode was a 1 cm × 1 cm T2 Cu electrode. The surface of the Cu electrode was polished first with 4000# sandpaper, and then, with a 1 μm diamond spray, cleaned with deionized water, activated with dilute acid, cleaned with deionized water, and then, blown dry. Linear sweep voltammetry (LSV) was performed at the open-circuit potential to −2.0 V with a scanning speed of 10 mV/s. The electrodes were tested using a linear voltammetry scanner. 

The Cu-Zn coating obtained by electroplating was etched using an etching solution to obtain nanoporous Cu. The temperature of the etching solution was 80 °C, and the etching solution consisted of 5.3 g of NH₄Cl, 12–13 mL of HCl (36–38 wt.%), and 79.7 g of distilled water.

The morphology of the Cu-Zn groups in the plating solution was observed using TEM (FEI, Hillsboro, OR, USA). The microstructures of the alloy particles, solder joints, and shear fracture surfaces were characterized using a focused ion beam/scanning electron microscope (FIB/SEM; FEI, Hillsboro, OR, USA) equipped with an energy-dispersive X-ray detector (EDX; EDAX, San Francisco, CA, USA)). The nanoporous Cu/Cu-Zn interface was sectioned using an FIB/SEM system to facilitate analysis and subsequently examined by TEM.

## 3. Results and Discussion

### 3.1. Analysis of the Electrodeposition Behavior of Cu-Zn

Metal ions and L-histidine, a polyhydroxy ligand, form heteroligand macromolecular compounds in plating solutions. The introduction of pectin into histidine significantly improves the stability of this macromolecule complex, owing to the effect of pectin on the supramolecular structure of the polymer matrix, with the use of heteroligands to help metal ions form more stable compounds [16]. According to the principle of hard and soft acids and bases, a ligand is more stable when it has more hydroxyl groups because the cation is more strongly coordinated to the hydroxyl group than to the carboxyl group. In addition, hydroxyl groups promote hydrogen bonding, which contributes to the stabilization of metal complexes; complexes based on hydroxyl groups are more stable than complexes based on individual amino acids. When low-ester pectin is added to histidine, some of its methyl-ester groups are converted to primary amides, which cross-link with divalent metal ions, such as Cu^2+^ and Zn^2+^, thereby promoting the stability of the resulting super-ligand.

ZnSO_4_ was added to CuSO_4_ at a molar ratio of 2:1 in a solution of L-histidine. Complex formation was confirmed by a positive potential shift of approximately 500 mV (Figure 1); here, the imidazole ring and carboxylic acid group are involved in the complexation of Cu^2+^ and Zn^2+^. We suggest that the addition of low-ester pectin to the plating solution results in the formation of a regular, curved, double-chain conformation due to the dimerization of the chain sequence and interchain chelation of cations at specific binding sites on each chain. This supposition was confirmed by LSV. L-Amino acids and their derivatives tend to form chelating rings with metal cations. The imidazole portion of histidine simultaneously acts as a proton donor, proton acceptor, and nucleophilic reagent; it can also exhibit reciprocal isomerism. The affinity of Zn^2+^ for histidine was similar to that of Cu^2+^; thus, the reduction of Cu and Cu-Zn alloys on Cu occurs through a transient nucleation process. 

The concentration of Cu complexes in the plating solution exerts a slight effect on the kinetics of Cu-Zn plating, as evidenced by the decrease in the cathodic current with decreasing Cu-complex concentration. However, the Cu-complex concentration has a significant effect on the thermodynamics of plating. Figure 1 shows that the deposition potentials of Zn and Cu shift negatively with the addition of the complex (pyrophosphate), thus suggesting that the addition of the complex during deposition is beneficial because it allows the reduction potential of Cu^2+^ to approach that of Zn^2+^; thus, the co-deposition of Cu and Zn occurs. TEM observations of the cathode reveal that Zn^2+^ and Cu^2+^ are preferentially reduced near the cathode to particles with a diameter of approximately 10 nm; this nucleation occurs as soon as the ions come into contact with the cathode, and the nucleated Cu-Zn nanoparticles are captured by the plating layer by means of “orientation attachment” [17,18]. The size of the plating layer increases as the surrounding nanoparticles are consumed. As plating progresses, dislocations gradually disappear, resulting in a more uniform contrast. Cu-Zn nanoparticles with different orientations can fuse with each other to form Cu_5_Zn_8_ or Cu_0.53_Zn_0.47_, which have similar orientations (Figure 2). However, dislocations occur on the inner face of the coating because the time and energy required to further adjust the crystal structure are insufficient. As the roughness of the coating increases, the dislocations generated by the merging of multiple particles and dispersed distribution of dislocations are normalized, merging into a few major dislocations, that is, grain boundaries. Thus, the grain size of the Cu-Zn coating is only approximately 1 μm [19,20,21].

The current first decreases, and then, increases owing to the charging of the electrode double layer, which is caused by the formation and growth of nuclei on the cathode surface. The increasing current peaks at a broad maximum owing to the increase in surface area caused by the three-dimensional growth of the metal on the nuclei. Subsequently, the current decreases on account of the increase in thickness of the diffusion layer. Finally, the current is maintained at a constant value. Nucleation can be categorized as transient or progressive. The slow growth of particles observed at a small number of activation sites during the initial stages of nucleation can be explained by a transient nucleation mechanism. Scharifker et al. [22,23] describe nucleation as follows:$$\frac{i^{2}(t)}{{i_{max}}^{2}}=\frac{1.2254}{t/t_{max}}{\left\{1-exp\left[2.3367(\frac{t}{t_{max}})\right]\right\}}^{2}$$
where $i_{max}$ and $t_{max}$ are the maximum peak current and time required to obtain the peak current in the time–current graph, respectively. The polarization curves of Cu, Zn, and Cu-Zn alloys deposited on the substrate using different complexing agents are shown in Figure 3. The reduction of Cu and Cu-Zn alloys on the Cu substrate exhibits the characteristics of transient nucleation. The deposition potential of the Cu-Zn alloys shifts toward the negative direction with increasing complex concentration. Thus, as plating proceeds, the composition of the plated eutectic alloy evolves from Cu_0.53_Zn_0.47_ to Cu_5_Zn_8_. In addition, stirring disrupts the deposition of Cu-Zn alloys. Specifically, when stirring is applied, a “sandwich structure” (Figure 4) in which the surface of the plated layer changes back to Cu_0.53_Zn_0.47_ instead of Cu_5_Zn_8_ forms. By comparing the process without stirring, we found that the addition of stirring makes the atomic deposition process more “chaotic”, that is, Cu^2+^ and Zn^2+^ tend to be deposited in a eutectic manner rather than independently. As shown in Figure 3, the independent deposition peak gradually disappears after applying stirring. As electroplating proceeds, the concentration of Cu^2+^ in the cathode area first decreases, and then, increases due to the presence of the sacrificial anode (H59 brass plate). Later, annealing of the coating helps the two phases to separate more clearly. In other words, the formation of the sandwich structure is due to the stirring behavior that promotes the deposition of eutectic alloy and the concentration change in Cu^2+^ (ratio relative to the concentration of Zn^2+^) brought by the sacrificial anode.

Based on the electron micrographs of the cross-section of the Cu-Zn plating layer close to the Cu substrate (Figure 4), the surface of the plating layer is composed of fine Cu_0.53_Zn_0.47_ grains, with a dense and smooth structure with a high Cu atom content. The Cu_5_Zn_8_ eutectic alloy is sandwiched between the Cu_0.53_Zn_0.47_ grains. The surface structure of the alloy precursor plays a crucial role in the ligament organization of the final foam material because a flat surface coating is essential to obtain robust ligaments. The flat and fine Cu_0.53_Zn_0.47_ grains provide numerous grain boundaries at the coating surface that enable the etching fluid to enter the Cu-Zn alloy. Meanwhile, the intermediate porous Cu_5_Zn_8_ alloy facilitates etching to obtain a sufficient number of nanopores. The Cu_0.53_Zn_0.47_ alloy close to the Cu substrate prevents the erosion of the Cu substrate by the corrosive fluid during etching.

### 3.2. Analysis of the Dealloying Behavior of Cu-Zn

Dealloying is a common etching process in which an alloy is “separated” by the selective dissolution of its most electrochemically active elements. This process results in the formation of a nanoporous sponge composed almost entirely of Cu. Although considerable attention has been paid to the morphological aspects of dealloying, the underlying physical mechanisms remain unclear. In this study, the structural interface of the foam was observed using AC-TEM.

The degree of dealloying increases with increasing dealloying time. During dealloying, Zn atoms in the surface layer are preferentially dissolved, and the remaining inert Cu atoms diffuse and accumulate without blocking the active atoms of the next layer, that is, Zn, from continuous dissolution. Consequently, a dealloyed etching channel is formed (Figure 5). This process is repeated, and the channels that continue to penetrate the material transform into nanoporous pores, eventually forming a nanoporous structure (Figure 6). As dealloying continues, the active components available for etching continue to diminish, and the inert component Cu begins to aggregate; thus, the Cu atoms are unable to form a homogeneous and continuous ligament structure at the surface layer. Owing to the reduction in surface energy, the Cu atoms undergo surface diffusion along the interface between the alloy and electrolyte, and the ligament size increases continuously, leading to the coarsening of the nanoporous structure. Because the surface-inert Cu atoms at the alloy/solution interface remain mobile, this coarsening process reduces the surface energy and results in a more stable structure [16,24]. From the perspective of thermodynamics, a dynamic solid–liquid interface forms during dealloying; in the case of long reaction times, the aggregation of Cu atoms and the dissolution and deposition of the smaller band structures in the nanoporous structure onto the coarser ligaments to form island-like particles continue. This process is equivalent to the Oswald ripening phenomenon within the thermodynamic dealloying system, which reaches a steady state with minimal energy. Smaller units continue to disappear, relatively larger units gradually expand, and the overall average size increases (up to a scale of about 200 nm). In summary, an appropriate dealloying time is crucial to obtain a desirable nanoporous structure (Figure 7) [25,26].

The accepted mechanism for the dealloying of single-phase solid-solution alloys is typically based on coherent surface diffusion, which maintains the grain structure and orientation of the starting alloy [21,27]. However, this process requires the starting and dealloyed materials to have the same crystal structure and similar lattice constants. However, excessive stresses at the dealloying front caused by changes in crystal structure or lattice constants are expected to alter the grain structure by triggering nucleation. Hence, we designed a sandwich-structured alloy precursor, Cu_0.53_Zn_0.47_/Cu_5_Zn_8_/Cu_0.53_Zn_0.47_, via electroplating; notably, the outermost layer of the precursor, Cu_0.53_Zn_0.47_, has the same crystal lattice, that is, a face-centered cubic structure, as the Cu atoms. Therefore, the etching behavior of this layer is initially based on coherent surface diffusion, and the obtained Cu crystals have the same orientation as the precursor (Cu_0.53_Zn_0.47_ eutectic alloy). By contrast, the etching of the intermediate Cu_5_Zn_8_ layer triggers changes in the crystal structure and lattice constants.

During the dealloying of the Cu_5_Zn_8_ alloy, Zn dissolves from surface areas such as steps. Cu atoms accumulate on both sides of the alloy, further preventing local dissolution. When approximately 10 monomolecular layers of the alloy are dissolved, dissolution ceases or is significantly delayed. Prior to the corrosion of the next layer, the Cu atoms spread and begin to accumulate into islands. Thus, instead of a uniformly diffused Cu layer, the coating surface consists of two distinct regions: pure Cu clusters on the locally passivated surface and plaques of unalloyed material exposed to the electrolyte (Figure 5). As the Zn atoms in these patches dissolve (Figure 7C,D), more Cu atoms are released to the coating surface. These atoms diffuse into the Cu clusters left by the dissolution of the previous layer so that the unalloyed material is increasingly exposed to the electrolyte. The dynamic equilibrium of adsorbed atoms is induced by their two-dimensional evaporation from the step edge to the grain boundary and subsequent recondensation. Because this process is coherent and the nucleation of Cu clusters is more likely to form on the step edges than at other locations, the rate decline is not significant and the increase in step edge sites promotes the overall reaction [28].

The rapid stripping of steps of Zn atoms in the Cu_5_Zn_8_ alloy leaves behind Cu atoms, which have the same local site occupancy as the whole Cu-Zn alloy. Therefore, a very strong driving force is required to coalesce Cu atoms with nearby Cu-rich clusters. The motion at the alloy–electrolyte interface can be mathematically described by the flux of diffusing atoms, $J_{s}$; the interfacial velocity perpendicular to itself, $\nu_{n}$; and the rate of concentration accumulation, all of which are interrelated and vary with the position on the interfacial curve. The flux of diffusing atoms can be described using the Cahn–Hilliard equation, which indicates that the normal velocity depends on the concentration of Cu atoms, *C*, and the local curvature through capillary effects, k. The time evolution of *C* is uniquely determined by local mass-conservation conditions [29].
$$\frac{\partial C}{\partial t}=\nu_{n}C_{0}-\nu_{n}kC-\nabla {\cdot J}_{s}$$
where $C_{0}$ is the concentration of bulk Cu. In this case, the thickness of the interfacial layer is constant along the interface rather than spatially varying in macroscopic diffusion length. The surface-aggregation process inherent in the Cahn–Hilliard form of $J_{s}$ is critical for pore formation. Thus, surface diffusion in steps produces an initially unstable interface, which, in turn, passivates rapidly before well-formed pores are formed. In this case, surface diffusion is driven by capillary forces (Figure 8). 

Observations of the FIB sections by TEM reveal the presence of capillary channels with diameters of approximately 10 nm between pores. Therefore, during etching, all remaining Cu atoms are transferred to the coating surface through capillary channels, where they are free to aggregate into islands. 

In other words, the formation of pores in nanoporous Cu is due to the chemical aggregation of inert Cu atoms into two-dimensional clusters during phase separation at the solid–electrolyte interface and the erosion of active Zn atoms to form capillary channels. The Cu atoms are then transferred to the surface by capillary action, and the surface area of the coating increases owing to etching. Together, these processes lead to changes in porosity according to the characteristic length scales predicted by the continuum models, ultimately leading to the formation of nanoporous Cu.

## 4. Conclusions

(1) The electrodeposition behavior of a Cu-Zn eutectic alloy precursor was investigated. The Cu_0.53_Zn_0.47_/Cu_5_Zn_8_/Cu_0.53_Zn_0.47_ composite precursor was obtained by adding macromolecules to the plating solution, followed by stirring. Cu_0.53_Zn_0.47_ had a flat surface and fine grains; thus, the nanoporous Cu ligaments obtained after etching were thick. Cu_5_Zn_8_ had a porous structure, which allowed for the sufficient etching of the inner regions of the precursor. The porous Cu foam obtained from this multilayer design had a strong surface layer and a porous inner layer.

(2) The dealloying behavior of Cu-Zn eutectic alloys was investigated. The dealloying of Cu_0.53_Zn_0.47_ was dominated by coherent surface diffusion, and the Cu crystals obtained following dealloying had the same crystal lattice and orientation as those of Cu_0.53_Zn_0.47_. The dealloying of Cu_0.53_Zn_0.47_ was dominated by capillary forces, and the Cu atoms obtained by dealloying were transported to the surface of the alloy, forming island-like clusters and undergoing nucleation growth due to capillary forces.

## Figures and Tables

**Figure 1 nanomaterials-14-00487-f001:**
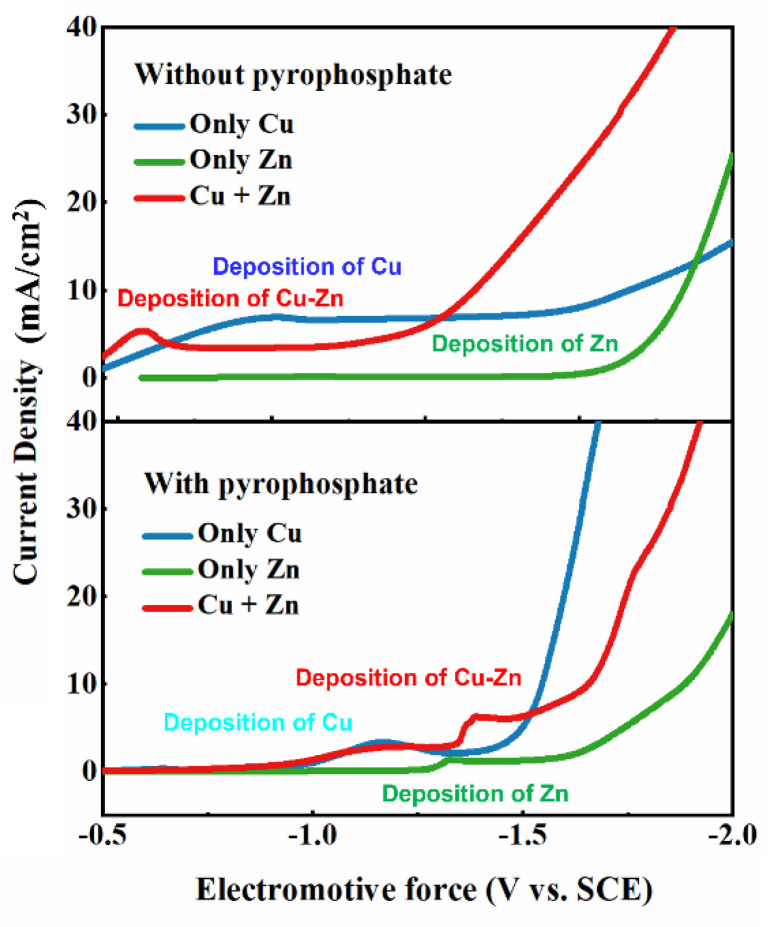
Polarization curves of different plating solutions with and without the addition of pyrophosphate.

**Figure 2 nanomaterials-14-00487-f002:**
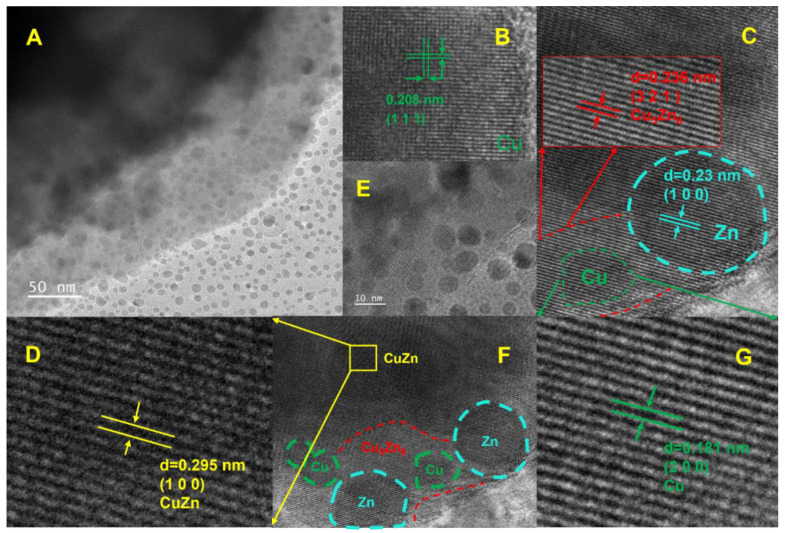
Co-deposition behavior of Cu and Zn during electroplating. (**A**) TEM images of the cathode surface. (**B**) High-resolution TEM (HRTEM) image of a Cu cluster. (**C**) HRTEM image of a Cu-Zn cluster. (**D**) HRTEM image of a Cu-Zn alloy. (**E**) HRTEM image of a Cu-Zn alloy and Cu-Zn cluster. (**F**) HRTEM image of a Cu-Zn alloy and Cu-Zn cluster. (**G**) HRTEM image of a Cu cluster.

**Figure 3 nanomaterials-14-00487-f003:**
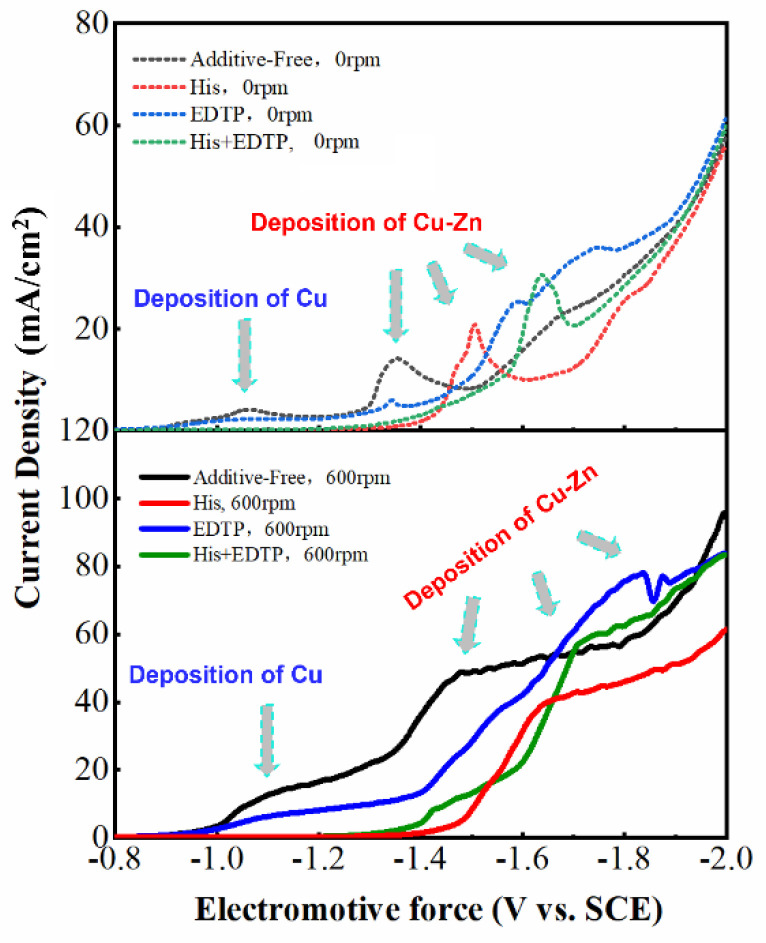
Polarization curves of Cu, Zn, and Cu-Zn alloys deposited on the substrate using different plating solutions with and without stirring (600 rpm).

**Figure 4 nanomaterials-14-00487-f004:**
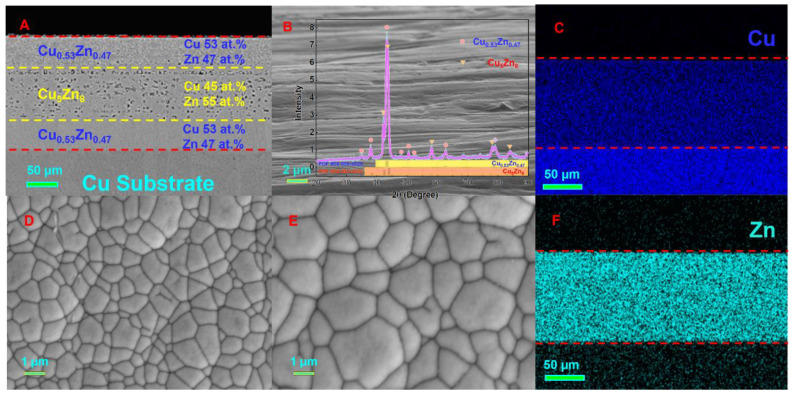
SEM images of the Cu-Zn eutectic alloys. (**A**) Sandwich structure of the eutectic alloys. (**B**) XRD scanning results of the eutectic alloys. (**C**) EDX results of the elemental distribution of Cu. (**D**) Surface topography of Cu_0.53_Zn_0.47_. (**E**) Grain size of Cu_0.53_Zn_0.47_. (**F**) EDX results of the elemental distribution of Zn.

**Figure 5 nanomaterials-14-00487-f005:**
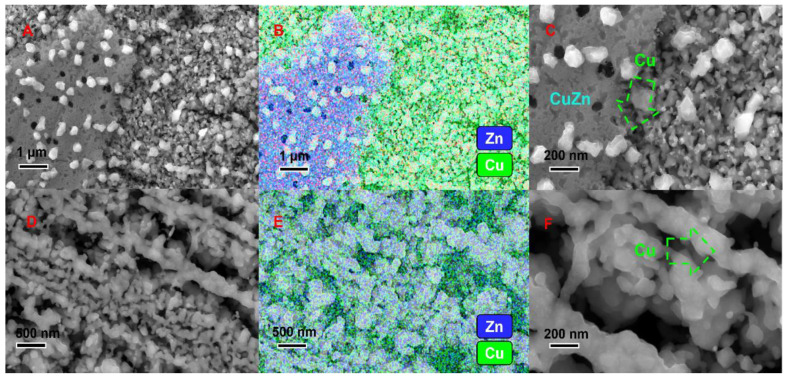
SEM images of the Cu-Zn eutectic alloy which is corroding. (**A**) SEM image of the Cu-Zn eutectic alloy which is corroding. (**B**) EDX results of mapping. (**C**) Corroded Cu-Zn alloy with residual Cu. (**D**) Cu foam organization remaining after dealloying of Cu-Zn alloys. (**E**) EDX results of Cu foam organization mapping. (**F**) SEM image of nanoporous copper.

**Figure 6 nanomaterials-14-00487-f006:**
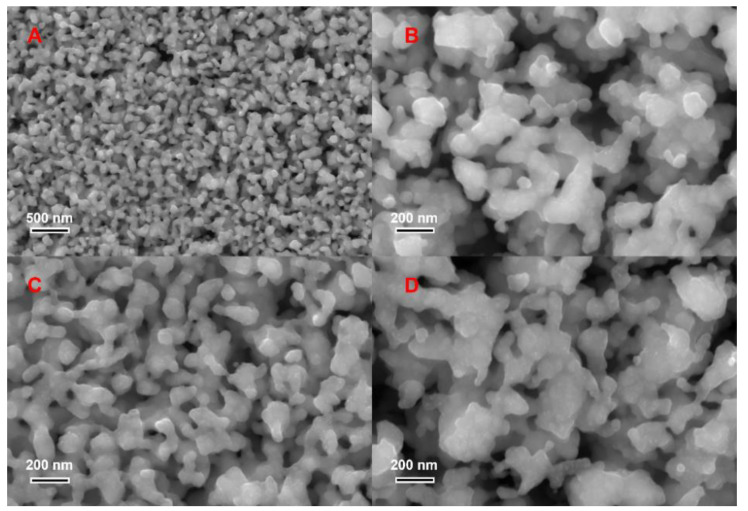
SEM images of nanoporous Cu after etching. (**A**) The microstructure of nanoporous Cu at low magnification. (**B**–**D**) The microstructure of nanoporous Cu at high magnification.

**Figure 7 nanomaterials-14-00487-f007:**
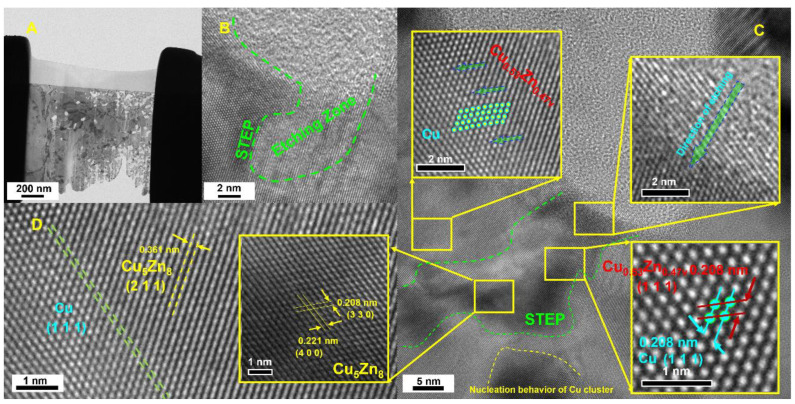
SA-TEM images of nanoporous Cu. (**A**) SA-TEM image of the FIB-sliced specimen. (**B**) SA-TEM image of the etching and etched zones. (**C**) SA-TEM image of the etching zone. (**D**) SA-TEM image of the Cu/Cu_5_Zn_8_ interface.

**Figure 8 nanomaterials-14-00487-f008:**
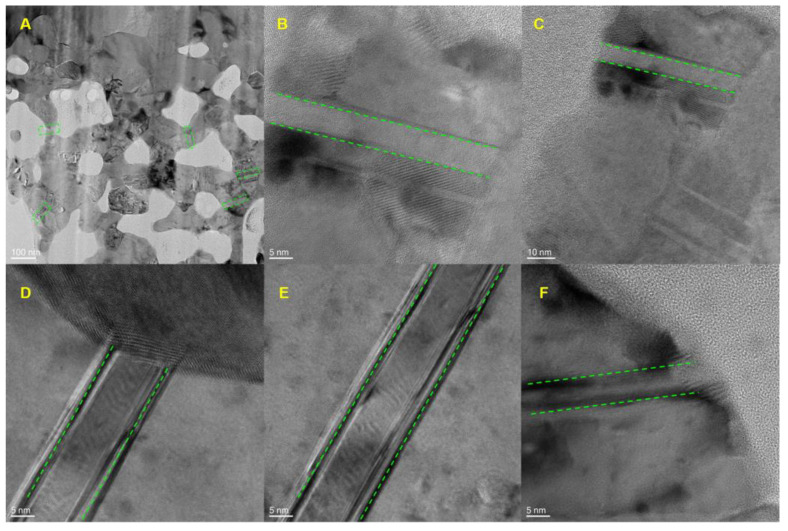
Capillary channels in nanoporous Cu. (**A**) TEM image of nanoporous Cu. (**B**–**F**) TEM images of the capillary channels.

## Data Availability

Data are contained within the article.

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
