# Peer review of "A Study of Electroplated Nanoporous Copper Using Aberration-Corrected Transmission Electron Microscopy"

_nanomaterials, 2024, doi:10.3390/nano14060487_

Round 1

Reviewer 1 Report

Comments and Suggestions for Authors

The main and interesting point of the manuscript is TEM analysis. It is done very precisely and have a great value. By this reason I suggest to rename the paper including the Case study of TEM.

The structure of paper is fine, but there a lot amount of typos, even in the chapter names (e.g. "20. Experimental").

Comments on the Quality of English Language

The English should be significantly corrected. Long sentences are difficult to understand.

Author Response

Comments 1: 
The main and interesting point of the manuscript is TEM analysis. It is done very precisely and have a great value. By this reason I suggest to rename the paper including the Case study of TEM.
Response 1: 
Thank you for your suggestion. TEM is indeed an important research method in this article. After discussion, we changed the original title from"A study of electroplated nanoporous copper: Deposition and Dealloying" to "A study of electroplated nanoporous copper by AC-TEM".

Comments 2: 
The structure of paper is fine, but there a lot amount of typos, even in the chapter names (e.g. "20. Experimental").
Response 2: 
We feel sorry for our carelessness. In our resubmitted manuscript, the typo is revised. We also checked our manuscript again and corrected these errors in the article. Some corrections are as follows:
1.  We’ve changed [20. Experimental] to [2. Experimental] (page 2, line 33)
2.  We’ve changed [where] to [Where] (page 5, line 18)
3.  We’ve changed [pyrophosphoric] to [potassium pyrophosphate] (page 2, line 39)

Reviewer 2 Report

Comments and Suggestions for Authors

The study reported by Jianqiang Wang et al succesfully obtained using a electroploting method a sandwich-structure Cu-Zn eutectic alloy precursor. The 3D porous structure were prepared using a delloying method and the obtained compound have been morphological characterized before and after dealloying.

After careful reading of the paper i recommended the publication of the study in Nanomaterial jurnal after performing the following suggestion.

1. From the introduction it is not clearly the applications of the obtained compounds

2. The references are a little bit too old. There are only one reference in the last year.

Author Response

Comments 1:

The study reported by Jianqiang Wang et al succesfully obtained using a electroploting method a sandwich-structure Cu-Zn eutectic alloy precursor. The 3D porous structure were prepared using a delloying method and the obtained compound have been morphological characterized before and after dealloying.

After careful reading of the paper i recommended the publication of the study in Nanomaterial jurnal after performing the following suggestion.

1. From the introduction it is not clearly the applications of the obtained compounds

Response 1:

We thank the reviewer for pointing this out. We have rewritten the introduction of the paper to clearly demonstrate the application of the Cu-Zn alloy layer. Briefly, the Cu-Zn alloy layer is the precursor alloy for preparing nanoporous Cu. In future research, we will transform the Cu-Zn alloy layer into a nanoporous Cu layer through dealloying method, which will then be used for interconnection of micro-joints during chip manufacturing.

Comments 2: The references are a little bit too old. There are only one reference in the last year.

Response 2:

While rewriting the introduction to the paper, we also updated the literature. Now, there are 5 references from this year and 4 references from last year. Specific references are as follows:

[1] Z.X. Cai, S. Bolar, Y. Ito,T. Fujita, Enhancing oxygen evolution reactions in nanoporous high-entropycatalysts using boron and phosphorus additives, Nanoscale (2024).

[2] Z.R. Chen, W. Zhao, Q. Liu,Y.F. Xu, Q.H. Wang, J.M. Lin, H.B. Wu, Janus Quasi-Solid Electrolyte Membraneswith Asymmetric Porous Structure for High-Performance Lithium-Metal Batteries,Nano-Micro Letters 16(1) (2024).

[3] T.Z. Jian, W.Q. Ma, J.G. Hou,J.P. Ma, X.H. Li, H.Y. Gao, C.X. Xu, H. Liu, Alloy/layer double hydroxideinterphasic synergy via nano-heterointerfacing for highly reversibleCO2 redox reaction in Li-CO2 batteries, Nano Research (2024).

[4] Z.H. Li, L. Feng, G.H. Huo, J.Y. Hao, C. Li, Z.Z. He, E.Sheremet, Y.D. Xu, J.X. Liu, Self-Assembly of Nanoporous ZIF-8-BasedSuperstructures for Robust Chemical Sensing of Solvent Vapors, Acs Applied NanoMaterials 7(3) (2024) 3479-3487

[6] R.A. Sosa, A. Antoniou, V. Smet, Ieee, Reliability and Failure Analysis of Chip-to-Substrate Cu-Pillar Interconnections with Nanoporous-Cu Caps, IEEE 73rd Electronic Components and Technology Conference (ECTC), Orlando, FL, 2023, pp. 318-323.

[8] S.-j. Han, S. Lee, K.-S.Jang, Epoxy-Based Copper (Cu) Sintering Pastes for Enhanced Bonding Strengthand Preventing Cu Oxidation after Sintering, Polymers 16(3) (2024).

[9] S. Thekkut, R.S.Sivasubramony, A. Raj, Y. Kawana, J. Assiedu, K. Mirpuri, N. Shahane, P.Thompson, P. Borgesen, Effective Constitutive Relations for Sintered NanoCopper Joints, Journal of Electronic Packaging 145(2) (2023).

[10] A. Rodrigues, A. Roshanghias, A Comparison Between Pressure-less and Pressure-assisted Cu Sintering for Cu Pillar Flip Chip Bonding, 2023 46th International Spring Seminar on Electronics Technology (ISSE)  (2023) 1-4.

[12] E. Castillo, M. Njuki, A.F. Pasha, N. Dimitrov, Copper-Based Nanomaterials for Fine-Pitch Interconnects in Microelectronics, Accounts of Chemical Research 56(12) (2023) 1384-1394.

Reviewer 3 Report

Comments and Suggestions for Authors

The research results presented in the manuscript are interesting and performed at a high level, the work is well written, well structured and presented. This did not raise any questions for me either from a chemical or thermodynamic point of view. The work presents interesting data on the coprecipitation of copper and zinc, which differ greatly in potential, which makes it possible to determine them separately. But the authors developed conditions that allowed them to co-deposit on the cathode surface. My only criticism was the lack of concentrations of reagents and pH of solutions in the text, on which the electrochemical parameters of the system can really depend.

Author Response

Comments 1: 
The research results presented in the manuscript are interesting and performed at a high level, the work is well written, well structured and presented. This did not raise any questions for me either from a chemical or thermodynamic point of view. The work presents interesting data on the coprecipitation of copper and zinc, which differ greatly in potential, which makes it possible to determine them separately. But the authors developed conditions that allowed them to co-deposit on the cathode surface. My only criticism was the lack of concentrations of reagents and pH of solutions in the text, on which the electrochemical parameters of the system can really depend.
Response 1: Thank you very much for your suggestion. The lack of reagent concentration and the pH of the solution have been added in the revised manuscript(page 2, line 39-41 ),as follows:
The complexing agent was 330 g/L potassium pyrophosphoric, the precursors were 12.5 g/L CuSOâ‚„·5Hâ‚‚O and 43.1 g/L ZnSOâ‚„·7Hâ‚‚O. The pH of the solution was adjusted to 12.5 by using KOH, and the temperature of the plating solution was 50 °C.

Reviewer 4 Report

Comments and Suggestions for Authors

The manuscript nanomaterials-2879405 by Wang, J.; Wang, J.; Lv, Z.; Zhang, L.; Chen, H.; Li, M. is an attempt by the authors to demonstrate a way for synthesizing nanoporous Cu films by a two-step approach involving a Cu-Zn alloy electrodeposition and subsequent de-alloying done by selective oxidative removal of Zn as less-noble component. The work is experimentally sound in general, yet many elements of the experimental activities are either not thoroughly explained or not presented in an objectively understandable perspective. Also, the motivation of doing the presented research is not clear to-the-point for a broader audience. In addition, the authors need to cite many references that are fundamental to this specific project but are largely missing in the current draft. In conclusion, the manuscript could probably be published eventually but before that the authors need to perform a moderate technical and major structural revision with specific guidelines as follows:
1. The authors need to give credit to other authors who traced the pathway to the synthesis of nanoporous Cu by de-alloying and cite at least the pioneer work (+) J. Hayes, A. Hodge, J. Biener, A. Hamza, and K. Sieradzki, J. Mater. Res., 21, 2611, (2006) for CuMn de-alloying.
2. The authors need to put their work in perspective of the state-of-the-art of current interconnection strategies for generation of small-scale (micro-) joints. Here, they need to talk about the common use of Cu nanopastes, and then work of others on nanoporous Cu (papers of A. Antoniou et al) and nanoporous CuSn (papers of N. Dimitrov et al, only one cited as ref. 8); some of these are listed below:
(+) K. Mohan, N. Shahane, R. Liu, V. Smet, and A. Antoniou, JOM, 70, 2192 (2018); (++) Mohan, K.; Shahane, N.; Raj, P. M.; Antoniou, A.; Smet, V.; Tummala, R., (++) Mohan, K.; Shahane, N.; Sosa, R.; Khan, S.; Raj, P. M.; Antoniou, A.; Smet, V.; Tummala, R., In 68th Electronic Components and Technology Conference (ECTC); IEEE: 2018; pp 301−307.
(+) Castillo, E.; Dimitrov, N. , J. Electrochem. Soc. 2021, 168, 062513. (++) Castillo, E.; Zhang, J.; Dimitrov, N. MRS Bull. 2022, 47, 913−925,
3. The authors need to make a bold distinction between their work and the already mentioned works of A. Antoniou et al and N. Dimitrov et al groups and convince the audience that they bring something new and unseen before with their present manuscript.
4. The authors need to do a better job in explaining the control "at will" of going back and forth from Cu0.53Zn0.47 to Cu5Sn8. Is this major compositional difference achieved only by turning on & off the stirring? If it is, then what is the underlaying phenomenology (physics; chemistry) behind this change? This approach should also be mentioned in the Experimental section.
5. The authors may want to explain how they did the cross-sectional imaging in Figure 4 as this procedure is usually challenging.
6. The authors must also address the question why in Figure 1A the kinetics of Cu(only) deposition seems to be sluggish as compared to the Cu+Zn deposition? Indeed, the most expected trend, in my opinion, is what we see in Figure 1b, where the Cu(only deposition starts first, then comes Cu+Zn, and finally goes Zn only. Please comment on that.

Comments on the Quality of English Language

Overall OK.

Author Response

The manuscript nanomaterials-2879405 by Wang, J.; Wang, J.; Lv, Z.; Zhang, L.; Chen, H.; Li, M. is an attempt by the authors to demonstrate a way for synthesizing nanoporous Cu films by a two-step approach involving a Cu-Zn alloy electrodeposition and subsequent de-alloying done by selective oxidative removal of Zn as less-noble component. The work is experimentally sound in general, yet many elements of the experimental activities are either not thoroughly explained or not presented in an objectively understandable perspective. Also, the motivation of doing the presented research is not clear to-the-point for a broader audience. In addition, the authors need to cite many references that are fundamental to this specific project but are largely missing in the current draft. In conclusion, the manuscript could probably be published eventually but before that the authors need to perform a moderate technical and major structural revision with specific guidelines as follows:

Comments 1:

The authors need to give credit to other authors who traced the pathway to the synthesis of nanoporous Cu by de-alloying and cite at least the pioneer work (+) J. Hayes, A. Hodge, J. Biener, A. Hamza, and K. Sieradzki, J. Mater. Res., 21, 2611, (2006) for CuMn de-alloying.

Response 1:

We sincerely appreciate the valuable comments. According to your suggestion, we rewrote the introduction part of the revised manuscript. We have also carefully read the reference you mentioned, which is very useful for our article. Its position in the revised manuscript is:

[page 2, line 3-4]: Hayes et al. prepared a precursor alloy by smelting and then dealloyed it to prepare uniform nanoporous Cu sheets [13]. (HAYES J R, HODGE A M, BIENER J, et al. Monolithic nanoporous copper by dealloying Mn–Cu[J]. Journal of Materials Research, 2006, 21(10): 2611-2616.)

Comments 2:

The authors need to put their work in perspective of the state-of-the-art of current interconnection strategies for generation of small-scale (micro-) joints. Here, they need to talk about the common use of Cu nanopastes, and then work of others on nanoporous Cu (papers of A. Antoniou et al) and nanoporous CuSn (papers of N. Dimitrov et al, only one cited as ref. 8); some of these are listed below:

(+) K. Mohan, N. Shahane, R. Liu, V. Smet, and A. Antoniou, JOM, 70, 2192 (2018); (++) Mohan, K.; Shahane, N.; Raj, P. M.; Antoniou, A.; Smet, V.; Tummala, R., (++) Mohan, K.; Shahane, N.; Sosa, R.; Khan, S.; Raj, P. M.; Antoniou, A.; Smet, V.; Tummala, R., In 68th Electronic Components and Technology Conference (ECTC); IEEE: 2018; pp 301−307.

(+) Castillo, E.; Dimitrov, N. , J. Electrochem. Soc. 2021, 168, 062513. (++) Castillo, E.; Zhang, J.; Dimitrov, N. MRS Bull. 2022, 47, 913−925,

Response 2:

We sincerely appreciate the valuable comments. We have also carefully read the reference you mentioned, which is very useful for our article. Its position in the revised manuscript is:

[page 1, line 33-36]: Nanometal materials are used in all-Cu interconnect technology because of their nanosize effect, which can significantly reduce the melting point [7]. (MOHAN K, SHAHANE N, LIU R, et al. A Review of Nanoporous Metals in Interconnects[J]. JOM, 2018, 70(10): 2192-2204.)

[page 2, line 10-11]: Mohan et al. prepared nano-porous copper by chemical dealloying, and its ligament size was about 60 nm [11]. (MOHAN K, SHAHANE N, SOSA R, et al. Demonstration of Patternable All-Cu Compliant Interconnections with Enhanced Manufacturability in Chip-to-Substrate Applications[C]//2018 IEEE 68th Electronic Components and Technology Conference (ECTC). San Diego, CA: IEEE, 2018: 301-307.)

[page 2, line 11-13]: Castillo et al. prepared nano-porous copper by electrochemical dealloying, and its liga-ment size is smaller, which is 20-33 nm [14]. (CASTILLO E, ZHANG J, DIMITROV N. All-electrochemical synthesis of tunable fine-structured nanoporous copper films[J]. MRS Bulletin, 2022, 47(9): 913-925.)

[page 2, line 11-13]: Dimitrov et al. believe that the content of Zn element in Cu-Zn coating should be above 60 at% to obtain good quality nanoporous copper [15]. (CASTILLO E, DIMITROV N. Electrodeposition of Zn-rich Cu x Zn (1−x) Films with Controlled Composition and Morphology[J]. Journal of The Electrochemical Society, 2021, 168(6): 062513.)

Comments 3:

The authors need to make a bold distinction between their work and the already mentioned works of A. Antoniou et al and N. Dimitrov et al groups and convince the audience that they bring something new and unseen before with their present manuscript.

Response 3:

We sincerely appreciate the valuable comments. We think it is the impact of different dealloying processes on the size of the ligament. The specific differences are as follows:

[page 2, line 10-11]: Mohan et al. prepared nano-porous copper by chemical dealloying, and its ligament size was about 60 nm [11].

[page 2, line 11-13]: Castillo et al. prepared nano-porous copper by electrochemical dealloying, and its liga-ment size is smaller, which is 20-33 nm [14].

Comments 4:  

The authors need to do a better job in explaining the control "at will" of going back and forth from Cu0.53Zn0.47 to Cu5Sn8. Is this major compositional difference achieved only by turning on & off the stirring? If it is, then what is the underlaying phenomenology (physics; chemistry) behind this change? This approach should also be mentioned in the Experimental section.

Response 4:

Thank you very much for your suggestion. This major compositional difference is achieved by turning on & off the stirring and annealing. By comparing the process without stirring, we found that the addition of stirring makes the atomic deposition process more "chaotic", that is, Cu2+ and Zn2+ tend to be deposited in a eutectic manner rather than independently. As shown in Figure 3, the independent deposition peak gradually disappeared after applying stirring. As electroplating proceeds, the concentration of Cu2+ in the cathode area first decreases and then increases due to the presence of the sacrificial anode (H59 brass plate). Later, annealing of the coating helps the two phases to separate more clearly. In other words, the formation of the sandwich structure is due to the stirring behavior that promotes the deposition of eutectic alloy, and the concentration change of Cu2+ (ratio relative to the concentration of Zn2+) brought by the sacrificial anode. This is also explained in the revised manuscript.(page 5,line 27-32;page ,line 1-4)

Comments 5:

The authors may want to explain how they did the cross-sectional imaging in Figure 4 as this procedure is usually challenging.

Response 5:

Thank you very much for your suggestion. The samples were first mounted in epoxy resin and then ground and polished to obtain cross-sections of the samples. The thickness of the coating has reached 180 μm, so it will be easier to prepare the cross-section of the coating.

Comments 6:

The authors must also address the question why in Figure 1A the kinetics of Cu(only) deposition seems to be sluggish as compared to the Cu+Zn deposition? Indeed, the most expected trend, in my opinion, is what we see in Figure 1b, where the Cu(only deposition starts first, then comes Cu+Zn, and finally goes Zn only. Please comment on that.

Response 6:

Thank you very much for your suggestion. We believe that when the plating solution contains two metal ions, there is an interaction between the two ions, resulting in a certain change in the deposition potential. In this study, the coupling effect between Zn ions and Cu ions caused a small positive shift in the deposition potential, so the current density of Cu+Zn was slightly higher than the current density of Cu at the beginning of deposition. Similar results were found in the article by CASTILLO et al. (Fig.1b).

CASTILLO E, DIMITROV N. Electrodeposition of Zn-rich Cu x Zn (1−x) Films with Controlled Composition and Morphology[J]. Journal of The Electrochemical Society, 2021, 168(6): 062513.

Round 2

Reviewer 4 Report

Comments and Suggestions for Authors

The authors did sufficient work in revising the paper and responding thoroughly and satisfactorily to all questions / comments. In my opinion, the paper is now suitable for publication. Congratulations on the good work!